# The Effectiveness of Isoplumbagin and Plumbagin in Regulating Amplitude, Gating Kinetics, and Voltage-Dependent Hysteresis of *erg*-mediated K^+^ Currents

**DOI:** 10.3390/biomedicines10040780

**Published:** 2022-03-27

**Authors:** Linyi Chen, Hsin-Yen Cho, Tzu-Hsien Chuang, Ting-Ling Ke, Sheng-Nan Wu

**Affiliations:** 1Institute of Molecular Medicine, National Tsing Hua University, Hsinchu 30013, Taiwan; lchen@life.nthu.edu.tw (L.C.); lynn103081035@gapp.nthu.edu.tw (T.-L.K.); 2Department of Medical Science, National Tsing Hua University, Hsinchu 30013, Taiwan; 3Department of Physiology, National Cheng Kung University Medical College, Tainan City 70101, Taiwan; lilyzhou861126@gmail.com (H.-Y.C.); fytg55qq@gmail.com (T.-H.C.); 4Institute of Basic Medical Sciences, National Cheng Kung University Medical College, Tainan City 70101, Taiwan

**Keywords:** isoplumbagin (5-hydroxy-3-methyl-1,4-naphthoquinone), plumbagin (5-hydroxy-2-methyl-1,4-naphthoquinone), *erg*-mediated K^+^ current, delayed-rectifier K^+^ current, voltage-gated Na^+^ current, current kinetics, voltage-dependent hysteresis, pituitary cell, Leydig cell

## Abstract

Isoplumbagin (isoPLB, 5-hydroxy-3-methyl-1,4-naphthoquinone), a naturally occurring quinone, has been observed to exercise anti-inflammatory, antimicrobial, and antineoplastic activities. Notably, whether and how isoPLB, plumbagin (PLB), or other related compounds impact transmembrane ionic currents is not entirely clear. In this study, during GH_3_-cell exposure to isoPLB, the peak and sustained components of an *erg* (*ether-à-go-go* related gene)-mediated K^+^ current (*I*_K(erg)_) evoked with long-lasting-step hyperpolarization were concentration-dependently decreased, with a concomitant increase in the decaying time constant of the deactivating current. The presence of isoPLB led to a differential reduction in the peak and sustained components of deactivating *I*_K(erg)_ with effective IC_50_ values of 18.3 and 2.4 μM, respectively, while the *K*_D_ value according to the minimum binding scheme was estimated to be 2.58 μM. Inhibition by isoPLB of *I*_K(erg)_ was not reversed by diazoxide; however, further addition of isoPLB, during the continued exposure to 4,4′-dithiopyridine, did not suppress *I*_K(erg)_ further. The recovery of *I*_K(erg)_ by a two-step voltage pulse with a geometric progression was slowed in the presence of isoPLB, and the decaying rate of *I*_K(erg)_ activated by the envelope-of-tail method was increased in its presence. The strength of the *I*_K(erg)_ hysteresis in response to an inverted isosceles-triangular ramp pulse was diminished by adding isoPLB. A mild inhibition of the delayed-rectifier K^+^ current (*I*_K(DR)_) produced by the presence of isoPLB was seen in GH_3_ cells, while minimal changes in the magnitude of the voltage-gated Na^+^ current were demonstrated in its presence. Moreover, the *I*_K(erg)_ identified in MA-10 Leydig tumor cells was blocked by adding isoPLB. Therefore, the effects of isoPLB or PLB on ionic currents (e.g., *I*_K(erg)_ and *I*_K(DR)_) demonstrated herein would be upstream of our previously reported perturbations on mitochondrial morphogenesis or respiration. Taken together, the perturbations of ionic currents by isoPLB or PLB demonstrated herein are likely to contribute to the underlying mechanism through which they, or other structurally similar compounds, result in adjustments in the functional activities of different neoplastic cells (e.g., GH_3_ and MA-10 cells), presuming that similar in vivo observations occur.

## 1. Introduction

Isoplumbagin (isoPLB, 5-hydroxy-3-methyl-1,4-naphthoquinone) is a naturally occurring quinone from *Lawsonia inermis* [1] or *Plumbago europarea* [2]. Similar to isoPLB, plumbagin (PLB, 5-hydroxy-2-methyl-1,4-naphthoquinone), another hydroxyl-1,4-napththoquinone, is an alkaloid obtained from the roots of the plants of the *Plumbago* genus. IsoPLB and PLB have recently been demonstrated to exert anti-neoplastic activity against an array of cancers, including oral or tongue squamous cell carcinoma, glioblastoma, non-small cell lung carcinoma, and breast, cervical, endometrial, pancreatic, and prostate cancers [3,4,5,6,7,8,9,10,11,12,13,14,15,16,17,18,19,20,21,22,23,24]. Nevertheless, PLB was shown to induce apoptotic changes in lung cancer or neuronal cells via either caspase-9 activation or the enhancement of mitochondrial-associated ROS [10,15,25,26]. PLB has been reported to modify the release of pituitary gonadotropin [27]. Previous studies have shown the ability of 2-mercaptophenyl-1,4-naphthoquinone, a naphthoquinone derivative, to overcome the elevation of intracellular Ca^2+^ in platelets caused by ADP and collagen [28], as well as to suppress the amplitude of Ca^2+^-activated and voltage-gated K^+^ currents in pituitary tumor (GH_3_) cells [29]. However, to our knowledge, detailed information is limited regarding the possible perturbations of isoPLB and its related compounds on different types of ionic currents, despite growing awareness of their ability to exert anti-neoplastic actions.

The *erg* (*ether-à-go-go* related gene)-mediated K^+^ current (*I*_K(erg)_), the necessary components for which are encoded by three different subfamilies of the *KCNH* gene, is able to generate the pore-forming α-subunit of *erg*-mediated K^+^ (i.e., K_erg_ or K_V_11) channels [30,31]. This type of K^+^ current is also thought to constitute a cloned counterpart of the rapidly activating delayed-rectifying K^+^ current residing in cardiac myocytes, where the *KCNH2* gene encodes the pore-forming α-subunit of the K_V_11.1 channels, commonly identified as hERG [31,32,33]. The *I*_K(erg)_ current is intrinsically present in neurons or varying types of excitable cells, such as neuroendocrine or endocrine cells. Of note, it can be engaged in the maintenance of resting potential and in modifications of the subthreshold excitability [34,35,36,37,38,39,40,41,42,43,44,45,46]. Previous work has also shown the effectiveness of the *I*_K(erg)_ magnitude in regulating the apoptosis and proliferation of many types of neoplastic cells [47,48,49,50,51,52,53,54]. However, whether or how the presence of isoPLB and PLB changes the magnitude of this type of K^+^ current is largely unclear.

Earlier reports have demonstrated that pituitary cells (e.g., GH_3_ lactotrophs), in addition to possessing voltage-gated K^+^ currents, can functionally express *I*_K(erg)_, the magnitude of which can perturb the functional activities of these cells [30,35,36,55,56,57,58,59,60,61,62]. This type of ionic current is unique, and is characterized by voltage-dependent activation and deactivation with an inwardly rectifying property. The magnitude of the current has been proposed to be a main determinant of the resting potential and, therefore, of membrane excitability [34,36,60]. Any modifications of this K_erg_-channel activity also have the propensity to alter the coupling of stimulus and secretion in these cells [30,36,55,60,63].

In view of the initiatives elaborated above, the objective of this work is to explore whether isoPLB, PLB, or other related compounds are capable of interacting with plasmalemmal ion channels to perturb the amplitude, gating, and voltage-dependent hysteresis (V_hys_) of membrane ionic currents, particularly the *erg*-mediated K^+^ current (*I*_K(erg)_). Our results demonstrate that the presence of isoPLB inhibits hyperpolarization-evoked *I*_K(erg)_ in a concentration-, time-, state-, and hysteresis-dependent manner in pituitary GH_3_ cells. The *I*_K(erg)_ intrinsically present in MA-10 Leydig tumor cells is also sensitive to inhibition by isoPLB.

## 2. Materials and Methods

### 2.1. Chemicals and Solutions Used in the Present Work

For the present study, isoplumbagin (isoPLB, 5-hydroxy-3-methyl-1,4-naphthoquinone, 8-hydroxy-2-methylnaphthalene-1,4-dione, 8-hydroxy-2-methyl-1,4-dihydronaphthalene-1,4-dione, C_11_H_8_O_3_) and plumbagin (PLB, 5-hydroxy-2-methyl-1,4-naphthoquinone, C_11_H_8_O_3_) were kindly provided by Dr. Linyi Chen (Institute of Molecular Medicine, National Tsing Hua University). The general procedures used for the purification of isoPLB and PLB have been reported in previous research [12].

Diazoxide, tetraethylammonium chloride (TEA), and tetrodotoxin (TTX) were supplied by Sigma-Aldrich (Merck, Tainan, Taiwan), PF-04856264 by Alomone (Jerusalem, Israel), and PD118057 by Tocris (Bristol, UK). Chlorotoxin was a gift provided by Professor Dr. Woei-Jer Chuang (Department of Biochemistry, National Cheng Kung University Medical College, Tainan, Taiwan), while azimilide was obtained from Procter and Gamble (Cincinnati, OH, USA). Unless stated otherwise, culture media (e.g., Ham’s F-12 medium), horse serum, fetal bovine or calf serum, L-glutamine, and trypsin/EDTA were obtained from HyClone^TM^ (Thermo Fisher; Genechain, Kaohsiung, Taiwan), while other chemicals, such as aspartic acid, CdCl_2_, CsCl, CsOH, EGTA, and HEPES, were of laboratory grade and supplied by Sigma-Aldrich. In this study, we used freshly deionized water prepared from a Milli-Q Ultrapure water purification system (Merck) in this work.

The ionic compositions of the external solution (i.e., HEPES-buffered normal Tyrode’s solution) were as follows (mM): NaCl 136.5, KCl 5.4, CaCl_2_ 1.8, MgCl_2_ 0.53, glucose 5.5, and HEPES-NaOH buffer 5.5; pH 7.4. To record the flow through K^+^ currents (e.g., *I*_K(erg)_ and *I*_K(DR)_), the patch pipette was filled with an internal solution containing the following (mM): K-aspartate 130, KCl 20, MgCl_2_ 1, KH_2_PO_4_ 1, Na_2_ATP 3, Na_2_GTP 0.1, EGTA 0.1, and HEPES-KOH buffer 5; pH 7.2. To measure *I*_K(erg)_ in GH_3_ or MA-10 cells, we replaced the bathing solution with a high-K^+^, Ca^2+^-free solution consisting of the following(mM): KCl 130, NaCl 10, MgCl_2_ 3, glucose 6, and HEPES-KOH buffer 10; pH 7.4. For *I*_Na_ measurements, we substituted the K^+^ ions inside the pipette internal solution for equimolar Cs^+^ ions, and the pH was then titrated to 7.2 by adding CsOH. The chemicals used to make these solutions were acquired from Sigma-Aldrich.

### 2.2. Cell Preparations

GH_3_ (a cell line from a rat anterior pituitary adenoma) pituitary tumor cells, supplied by the Bioresources Collection and Research Center ([BCRC-60016]; Hsinchu, Taiwan), were grown as monolayer cultures in plastic dishes containing Ham’s F-12 nutrient media (HyClone^TM^) supplemented with 15% horse serum, 2.5% fetal calf serum and 2 mM L-glutamine [55,56,59]. This cell line, originally derived from the American Type Culture Collection (ATCC^®^ [CCL-82.1TM]; Manassas, VA, USA), has been established as a model of electrically excitable cells in electrophysiology and pharmacology studies [29,35,58,63,64,65,66,67,68,69]. It was confirmed that this cell line can continually secrete prolactin. The MA-10 cell line, which was originally derived from a mouse Leydig tumor, was maintained in Waymouth nutrient media (HyClone^TM^,Thermo Fisher, Tainan, Taiwan) supplemented with 10% fetal bovine serum [68]. GH_3_ or MA-10 cells were cultured in a water-saturated environment of CO_2_/air (1:19). The experiments were conducted 5 or 6 days after cells underwent subculture (60–80% confluence).

### 2.3. Patch-Clamp Recordings: Electrophysiological Measurements

Shortly before each experiment, we gently dispersed GH_3_ or MA-10 cells with a 1% trypsin-EDTA solution, and an aliquot of suspension containing clumps of cells was transferred to a home-made chamber affixed to the working stage of a DM-IL inverted microscope (Leica; Major Instruments, Kaohsiung, Taiwan). We bathed cells at room temperature (20–25 °C) in HEPES-buffered normal Tyrode’s solution containing 1.8 mM CaCl_2_, the composition of which is described above. The pipettes that we used were prepared from Kimax^®^-51 borosilicate capillaries (#34500; Dogger, Tainan, Taiwan) by using either a vertical PP-83 puller (Narishige; Taiwan Instrument, Tainan, Taiwan) or a horizontal P-97 puller (Sutter, Novato, CA, USA), and their tips were fire-polished with an MF-83 microforge (Narishige). For efficient recordings, we chose electrodes with tip resistances of 3–5 MΩ, as they were filled with different internal solutions as stated above. Macroscopic ionic currents passing through the whole cell (i.e., whole-cell mode) with a modified patch-clamp technique were measured using either an RK-400 (Bio-Logic, Claix, France) or an Axopatch^TM^-200B (Molecular Devices^®^; Bestogen, New Taipei City, Taiwan) patch amplifier [57,66,68]. All potentials were corrected for liquid junction potentials that would arise at the pipette tip when the composition of the internal solution differed from that in the bath. The compounds tested were either applied through perfusion or added to the bath in order to achieve the final concentration indicated. In order to record the *I*_K(erg)_, we voltage-clamped the examined cell at a holding potential of −10 mV before a long-lasting hyperpolarizing step was delivered.

### 2.4. Data Recordings

Data acquisition was performed using either pCLAMP 10.6 (Molecular Devices), LabChart 7.1 (PowerLab; ADInstruments, Kuoyang, New Taipei City, Taiwan), or Hantek 6022BL (Qingdao, China). Current signals were low-pass filtered at 1 or 3 kHz. Analog signals were monitored and stored online at 5–10 kHz via analog-to-digital (AD) conversion in an ASUSPRO-BU401LG laptop computer (ASUS, Tainan, Taiwan), which was equipped with a low-noise Digidata^®^ 1440A device (Molecular Devices). With digital-to-analog (DA) conversion, the various voltage waveforms were specifically designed to determine the steady-state current-voltage (*I–V*) relationship, the recovery of the current block with a two-step voltage protocol in a geometric progression, or the V_hys_ of the ionic currents specified (e.g., *I*_K(erg)_). The data digitally acquired during the experiments were subsequently analyzed by different analytical tools, such as OriginPro^®^ 2021 (OriginLab; Scientific Formosa, Kaohsiung, Taiwan), LabChart 7.1 (ADInstruments), and custom-built macros in Excel^®^ 2021 spreadsheets under Windows 10 (Microsoft, Redmond, WA, USA).

### 2.5. Whole-Cell Data Analyses

To assess the percentage inhibition of isoPLB on the *I*_K(erg)_ amplitude, each tested cell was 1-s hyperpolarized from −10 to −90 mV, and the current magnitudes were measured at the beginning (i.e., peak amplitude) or end-pulse (i.e., sustained amplitude) of the voltage pulse during cell exposure to different isoPLB concentrations and then compared. The concentration of isoPLB required to suppress 50% of the peak or sustained component in the current amplitude (Figure 1C) was determined using a Hill function:y=Emax{1+(IC50nH/[C]nH)}
where *y* = percentage inhibition (%); [*C*] = the concentration of isoPLB used; *n_H_* = the Hill coefficient; *IC*_50_ = half-maximal inhibition; *E*_max_ = isoPLB-induced maximal inhibition of *I*_K(erg)_ (peak or sustained component).

The inhibitory effect of isoPLB on *I*_K(erg)_ seen in GH_3_ cells apparently occurred in a time-dependent manner and it can reasonably be explained by a state-dependent blocker which can preferentially bind to the open state (or conformation) of the K_erg_ channel. As such, from a simplified assumption, the first-order reaction scheme in which the isoPLB molecule interacts with the K_erg_ channel is given as follows:$$C\;\begin{array}{c} \alpha \\ ⇄\\ \beta \end{array}\;O\;\begin{array}{c} k_{+1}^{*}\cdot \left[isoPLB\right]\\ ⇄\\ k_{-1} \end{array}\;O\cdot \left[isoPLB\right]$$
where [isoPLB] indicates the isoPLB concentration given, α or β is the voltage-dependent rate constant needed for the opening and closing of the K_erg_ channels, respectively, and, *k*_+1_* or *k*_−1_ respectively represents the forward (on or bound) or backward (off or un-bound) rate constants of isoPLB binding, respectively. It means that the value of *k*_+1_*·[*isoPLB*] changes as a function of the isoPLB concentration, while *k*_−1_ is independent of the isoPLB concentration per second. C, O, or O·[*isoPLB*] shown in each term denote the closed (or resting), open, or open-[*isoPLB*] state, respectively.

The forward and reverse rate constants (i.e., *k*_+1_* and *k*_−1_) were determined from the time constants of the current decay changed upon step hyperpolarization from −10 to −90 mV with a duration of 1 s. The deactivation time constant (τ_deact_) of the resulting current was satisfactorily estimated by fitting the decaying trajectory of each current trace with a single exponential. The evolving rate constants (1/τ_deact_) would be then evaluated using the following equation:1Δτ=k+1*×[isoPLB]+k−1

In this linear relationship, *k*_+1_* or *k*_−1_ were respectively obtained either from the slope (i.e., ∆(1/∆*τ*)/∆([*isoPLB*]) or from the *y*-axis intercept (i.e., y intercept) at [*isoPLB*] = 0 of the linear regression into which the relationship of the 1/∆*τ* value versus the isoPLB concentrations was interpolated. ∆*τ* indicates the difference in the τ_deact_ value obtained in situations where the τ_deact_ values during exposure to different isoPLB concentrations (1–50 μM) were subtracted from those in the presence of 100 μM isoPLB (Figure 1B). A measure of the dissociation constant (*K*_D_), which is equal to *k*_−1_ divided by *k*_+1_* can thereby be calculated.

### 2.6. Curve Fitting Approximations and Statistical Analyses

The curve-fitting (linear or nonlinear) to experimental data acquired in this work was conducted with the goodness-of-fit using different methods, such as pCLAMP 10.7 (Molecular Devices), OriginPro 2021 (OriginLab), and Microsoft Excel^®^ 2021 (i.e., “Solver” add-in) under Microsoft Office 365 (Redmond, WA, USA). The whole-cell data are presented as mean ± standard error of the mean (SEM) with the size of experimental observations (n) indicative of cell numbers from which the results were taken, and error bars are plotted as SEM. The data distribution was found to satisfy the tests for normality. The Student’s *t*-test (paired or unpaired) between two groups was first used for the statistical analyses. When the differences among different groups needed to be testified, we further conducted an analysis of variance (ANOVA-1 or ANOVA-2), which was then followed by a post-hoc Fisher’s least-significance different test for multiple-range comparisons. Probability with *p* < 0.05 was considered to indicate statistical difference, unless noted otherwise.

## 3. Results

### 3.1. Effect of IsoPLB on I_K(erg)_ Recorded from GH_3_ Cells

In the initial stage of the measurements, we used the whole-cell configuration of the patch-clamp technique to evaluate if the presence of isoPLB leads to any perturbations on ionic currents (e.g., *I*_K(erg)_) residing in GH_3_ cells. In order to measure the current flowing through *I*_K(erg)_, we placed the cells to be bathed in a high-K^+^, Ca^2+^-free external solution which contained 1 μM tetrodotoxin (TTX) and 0.5 mM CdCl_2_, and we then backfilled the recording pipette using a K^+^-enriched (145 mM) internal solution. TTX or CdCl_2_ was used to suppress the magnitude of voltage-gated Na^+^ or Ca^2+^ currents [46,69], respectively, in order to avoid any interference with measurements of *I*_K(erg)_. As the whole-cell mode was established, we voltage-clamped the tested cell at the potential of −10 mV in voltage clamp mode and the hyperpolarizing step was thereafter applied (1 s in duration) to −90 mV for the elicitation of deactivating *I*_K(erg)_ with a slowly decaying time course, as demonstrated previously [46,57,58,59,67].

As illustrated in Figure 1A, one minute after cells were exposed to isoPLB at a concentration of 1 or 3 μM, the amplitude of *I*_K(erg)_ evoked by such long-lasting hyperpolarizing step progressively became decreased, and the decaying rate of the evolving current concurrently was reduced. For example, cell exposure to 3 μM isoPLB significantly decreased the peak and sustained components of *I*_K(erg)_ to 99 ± 11 pA (*n* = 8, *p* < 0.05) and 48 ± 8 pA (*n* = 8, *p* < 0.05) from the control values of 147 ± 22 pA and 96 ± 11 pA (*n* = 8), respectively. Meanwhile, under exposure to 3 μM isoPLB, the value for the deactivation time constant (τ_deact_) of *I*_K(erg)_ was also lengthened from 101 ± 18 to 287 ± 25 ms (*n* = 8, *p* < 0.05). After the washout of isoPLB, the peak and sustained amplitudes of *I*_K(erg)_ evoked by membrane hyperpolarization were returned to 142 ± 21 pA (*n* = 7) and 92 ± 9 pA (*n* = 7), respectively. Likewise, similar results were obtained during cell exposure to 3 μM PLB in seven different examined cells.

It was noted that the observed effect of isoPLB on *I*_K(erg)_ could not only emerge instantaneously, but could also emerge in a time- and concentration-dependent fashion. A linear relationship between the 1/Δτ_deact_ value and the isoPLB concentration is therefore illustrated in Figure 1B. The forward (*k*_+1_*) or backward (*k*_−1_) rate constant derived from the minimum binding and unbinding scheme were thereafter estimated to be 0.39 s^−1^μM^−1^ and 1.01 s^−1^, respectively; consequently, the value of the dissociation constant (K_D_ = *k*_−1_/*k*_+1_*) was optimally calculated to be 2.58 μM. This value was noticed to be similar to the IC_50_ value (indicated in Figure 1C) required for an isoPLB-induced block of sustained *I*_K(erg)_ measured at the end of the hyperpolarizing step.

Figure 1C illustrates that cell exposure to isoPLB in the bath can concentration-dependently depress the amplitude of the peak or sustained *I*_K(erg)_ evoked in response to the long-lasting hyperpolarizing step. According to the modified Hill equations detailed under Materials and Methods, the IC_50_ values needed for the isoPLB-mediated inhibitory effects on peak and sustained *I*_K(erg)_ were estimated to be 18.3 or 2.4 μM, respectively. The results reflect that isoPLB exercises a depressant action on the peak and sustained *I*_K(erg)_ present in GH_3_ cells and that this compound tends to be selective for sustained over peak *I*_K(erg)_ evoked by the hyperpolarizing step.

### 3.2. Steady-State I-V Relationship of Peak and Sustained I_K(erg)_ Caused by IsoPLB

We continued to examine if isoPLB-mediated changes in *I*_K(erg)_ amplitude can occur at different levels of command voltage pulse. In these experiments, we held the tested cell at −10 mV and a set of voltage steps ranging between −100 and 0 mV with a duration of 1 s was imposed. Figure 2 illustrates the average *I–V* relationship of the peak (black symbols) and sustained (red symbols) components of *I*_K(erg)_ with or without cell exposure to 3 μM isoPLB. Likewise, one minute after cell exposure to 3 μM isoPLB, the whole-cell conductance of peak *I*_K(erg)_ measured between −100 and −60 mV was significantly reduced to 2.12 ± 0.08 nS (*n* = 8, *p* < 0.05) from a control value of 2.81 ± 0.09 nS (*n* = 8). The results indicate that the application of this compound (3 μM) can effectively suppress the peak and sustained components of *I*_K(erg)_ measured throughout the entire voltage range imposed.

### 3.3. Comparisons among Effect of IsoPLB, PLB, 4,4′-Dithiodipyridine (DTDP), Chlorotoxin, Azimilide, IsoPLB plus Diazoxide, and DTDP plus IsoPLB on Hyperpolarization-Evoked I_K(erg)_ in GH_3_ Cells

The effects of isoPLB, PLB, DTDP, chlorotoxin, azimilide, isoPLB plus diazoxide, and DTDP plus isoPLB on the amplitude of *I*_K(erg)_ in GH_3_ ells were further assessed and compared. As demonstrated in Figure 3, the application of DTDP (10 μM) or azimilide (10 μM) was able to suppress *I*_K(erg)_ amplitude, while chlorotoxin, a blocker of Cl^−^ channels [70], was ineffective at inhibiting it. Azimilide is reported to be an inhibitor of *I*_K(erg)_ [67,71], and DTDP is a lipophilic sulfhydryl oxidizing agent [72,73]. Moreover, we found that with continued exposure to 3 μM isoPLB, the subsequent addition of diazoxide was noticed to have little or no effect on the isoPLB-mediated block of *I*_K(erg)_ in these cells. Diazoxide was shown to enhance the activity of ATP-sensitive K^+^ (K_ATP_) channels [74,75]. The subsequent application of 3 μM isoPLB, while still in the continued presence of 10 μM DTDP, did not suppress *I*_K(erg)_ further. These results suggest that isoPLB could share a similar mechanism of action to DTDP with respect to their block of *I*_K(erg)_ observed in GH_3_ cells, although neither the activity of K_ATP_ nor Cl^−^ channels residing in these cells is predominantly linked to isoPLB-induced block of *I*_K(erg)_.

### 3.4. Slowing in Recovery from I_K(erg)_ Block Caused by IsoPLB in GH_3_ Cells

We continued to explore if the presence of isoPLB could alter the recovery from the block of *I*_K(erg)_. In this set of experiments, we measured the recovery from the current block by using a standard gapped pulse protocol in which the interpulse interval varies with a geometric progression. The ratio of the amplitude of the second and first peak *I*_K(erg)_ (i.e., relative amplitude) evoked upon a 1-s hyperpolarizing pulse to −100 mV from a holding potential of −10 mV was imposed as a measure of the degree of recovery from the *I*_K(erg)_ block. The results appearing in Figure 4 show that, in the control period (i.e., absence of isoPLB), the peak amplitude of *I*_K(erg)_ was fully restored from the block when the interpulse interval was set at 1 s or above. Moreover, the time course of recovery from the current block achieved without and without cell exposure to 3 μM isoPLB was satisfactorily fitted by a single exponential, and time constants of 8.2 ± 2.2 (*n* = 8) and 22.4 ± 5.1 ms (*n* = 8) were derived, respectively. As such, the experimental results lead us to reflect that the presence of isoPLB produces an appreciable lengthening in the recovery from the block of peak *I*_K(erg)_ seen in GH_3_ cells.

### 3.5. Modification by IsoPLB of the Time Course of I_K(erg)_ Evoked by the Envelope-of-Tail Test

Earlier studies have demonstrated that there was a time-dependent change in *I*_K(erg)_ activation during the envelope-of-tail test [76,77]. In this set of envelope-of-tail experiments, we evoked an inward deactivating and an activating *I*_K(erg)_ elicited by varying durations of hyperpolarizing step (from 4 ms to 8.192 s with a geometric progression) to −90 mV from −10 mV. As can be seen in Figure 5, the relationship of the relative amplitude versus the pulse duration with or without the application of 10 μM isoPLB was established. In keeping with previous observations [76], the envelope-of-tail test tailored for the elicitation of *I*_K(erg)_ in this study was observed to reveal a time-dependent exponential decay in the ratio of the relative amplitude (i.e., *I*_act_/*I*_deac_) evoked during the durations between 4 ms and 8.192 s in a geometric progression, with or without the application of this compound (10 μM). With GH_3_-cell exposure to 10 μM isoPLB, the time constant of *I*_K(erg)_ activated by the envelope-of-tail method was significantly decreased to 21.2 ± 1.2 ms (*n* = 7, *p* < 0.05) from a control value of 44.3 ± 2.1. It is conceivable from the present observations, therefore, that isoPLB addition can produce an evident shortening in the time course of *I*_K(erg)_ activation evoked with the envelope-of-tail voltage protocol observed in GH_3_ cells.

### 3.6. Modification by IsoPLB of the V_hys_ Behavior of I_K(erg)_ Activated by Inverted Double Ramp Voltage (V_ramp_)

Earlier reports have demonstrated the capability of *I*_K(erg)_’s V_hys_ strength to perturb either various patterns of bursting, firing, or action potential configurations intrinsically in different types of excitable cells [31,36,37,38,39,40,41,42,43,44,46]. Therefore, we next assessed whether and how the exposure of isoPLB could modify the V_hys_ strength of the *I*_K(erg)_ evoked with double (i.e., inverted isosceles-triangular) V_ramp_. In this series of experiments, we voltage-clamped the tested cell at −10 mV, and through D/A conversion, a downward (forward or descending) ramp from −0 to −100 mV followed by an upward (backward or ascending) limb back to 0 mV (i.e., inverted isosceles-triangular V_ramp_) was then delivered (Figure 6). Under such a maneuver, the V_hys_ of *I*_K(erg)_ in response to such a double V_ramp_ was clearly observable; that is, the current magnitude evoked during the downward and upward limb of V_ramp_ (i.e., the instantaneous *I–V* relationship of *I*_K(erg)_) was highly distinguishable and the direction of *I*_K(erg)_ over time was noticed to display a clockwise direction. For example, during the control period (i.e., absence of isoPLB), the *I*_K(erg)_ amplitude taken at −80 mV during the downward ramp of double V_ramp_ was 423 ± 19 pA (*n* = 8), a value that was much greater than that measured at the same level of voltage during the upward end (45 ± 6 pA; *n* = 8, *p* < 0.01). As the V_ramp_ speed became slowed, the V_hys_ strength activated was progressively reduced.

Of additional interest, during cell exposure to 3 or 10 μM isoPLB, the evolving V_hys_ strength responding to an inverted double V_ramp_ was robustly lessened as the isoPLB concentration was raised (Figure 6). For example, as the duration of the double V_ramp_ delivered was set at 1.2 s with a ramp speed of ±183.3 mV/s, the value of ∆area (i.e., the difference in the area encircled by the curve in the downward and upward direction) for V_hys_ in the control period was calculated to be 19.5 ± 2.6 mV·nA (*n* = 8), while the value of ∆area in the presence of 3 or 10 μM isoPLB was measurably reduced to 10.1 ± 1.4 mV·nA (*n* = 8, *p* < 0.05) or 6.2 ± 0.8 mV·nA (*n* = 8, *p* < 0.05), respectively. Additionally, with cell exposure to isoPLB (10 μM), the further addition of PD118057 (10 μM) reversed the ∆area value to 12.3 ± 1.6 mV·nA (*n* = 8, *p* < 0.05). PD118057 has been reported to enhance *I*_K(erg)_ [78].

### 3.7. Mild Inhibitory Effect on Delayed-Rectifier K^+^ Current (I_K(DR)_) Produced by IsoPLB

Another set of experiments was undertaken to determine if different types of K^+^ currents (e.g., *I*_K(DR)_) in GH_3_ cells can be modified by the presence of isoPLB. We bathed cells in a Ca^2+^-free Tyrode’s solution which contained 1 μM TTX and 0.5 mM CdCl_2_, and the electrode was filled up with a K^+^-enriched solution. As isoPLB at a concentration of 1 μM was applied, no appreciable change in *I*_K(DR)_ amplitude elicited by the step depolarization to +50 mV from a holding potential of −50 mV was demonstrated. However, one minute of exposing cells to isoPLB (10 μM) decreased the *I*_K(DR)_ amplitude evoked by membrane depolarization. The average *I–V* relationship of the *I*_K(DR)_ obtained during the control period (i.e., absence of isoPLB) or cell exposure to isoPLB (10 μM) was established and is then illustrated in Figure 7A,B. For example, with cell exposure to 10 μM isoPLB, as the tested cell was 1-s depolarized from −50 to +50 mV with a duration of 1 s, the *I*_K(DR)_ measured at the end of the voltage pulse was significantly decreased to 545 ± 78 pA (*n* = 7, *p* < 0.05) from a control value of 690 ± 121 pA (*n* = 7). As isoPLB was removed, the current amplitude returned to 687 ± 119 pA (*n* = 7). Likewise, in the presence of 10 μM isoPLB, the whole-cell conductance of *I*_K(DR)_ measured at the potentials ranging from +10 to +50 mV was also reduced to 7.67 ± 0.21 nS (*n* = 7, *p* < 0.05) from a control value of 8.13 ± 0.23 nS (*n* = 7). These results prompt us to reflect that the presence of isoPLB (10 μM) yields a mild inhibitory effect on the *I*_K(DR)_ obtained in GH_3_ cells.

### 3.8. Inability of IsoPLB to Alter Voltage-Gated Na^+^ Current (I_Na_) in GH_3_ Cells

Some of the compounds which have been revealed to block *I*_K(erg)_ have been noticed to be engaged in altering the magnitude and gating kinetics of *I*_Na_ [33,57]. We therefore wanted to investigate whether isoPLB has any effects on the *I*_Na_ residing in GH_3_ cells. Here, we bathed cells in a Ca^2+^-free Tyrode’s solution, in which 10 mM TEA and 0.5 mM CdCl_2_ were included. The pipettes used were filled with a Cs^+^-containing solution, the composition of which was stated under Materials and Methods. The tested cell was depolarized from −100 to −10 mV with a duration of 30 ms to evoke *I*_Na_ with a fast activating and inactivation time course during a rapid depolarizing command pulse [33,69,79]. With the exposure to 10 μM isoPLB, the peak amplitude of the *I*_Na_ remained unaffected (Figure 8A,B). The presence of 10 μM PLB also had little or no effect on the *I*_Na_. However, during continued exposure to isoPLB, the subsequent addition of either PF-04856264 (3 μM) or TTX (1 μM) was active in suppressing the *I*_Na_ amplitude. PF-04856264 has been previously reported to be a synthetic blocker of Na_V_1.7 channels [80]. A summary bar graph demonstrating the effectiveness of isoPLB, PLB, isoPLB plus PF-04856264, and isoPLB plus TTX in altering *I*_K(erg)_ is illustrated in Figure 8B. Unlike *I*_K(erg)_, it thus appears that the *I*_Na_ evoked during rapid membrane depolarization is refractory to modification by the presence of isoPLB or PLB.

### 3.9. Effect of IsoPLB on I_K(erg)_ in MA-10 Leydig Tumor Cells

IsoPLB has been since reported to exercise anti-neoplastic actions in an array of cancer cells [3,47,48,49,51,52,53,54]. Therefore, in a final set of experiments, we further wanted to test if any modifications of *I*_K(erg)_ in other types of neoplastic cells (e.g., MA-10 cells) could be induced by the presence of isoPLB. The experimental protocol applied was similar to that used above in GH_3_ cells. The application of isoPLB drastically reduced the peak amplitude of the *I*_K(erg)_ measured over the entire voltage ranges examined. The average *I–V* relationship of peak *I*_K(erg)_ with or without exposure to 3 μM isoPLB was constructed and is hence illustrated in Figure 9. For example, cell exposure to isoPLB at a concentration of 3 μM significantly decreased the peak *I*_K(erg)_ measured at −100 mV to 318 ± 38 pA (*n* = 7, *p* < 0.05) from a control value of 615 ± 45 pA (*n* = 7). After isoPLB was removed, the current amplitude returned to 608 ± 41 pA (*n* = 7). It is evident from the results that, similar to that seen above in GH_3_ cells, the sizable *I*_K(erg)_ identified in MA-10 cells was sensitive to inhibition by the isoPLB presence, although MA-10 cells were not found to display the presence of *I*_Na_.

## 4. Discussion

The important findings shown in this study are that the presence of isoPLB can regulate the magnitude, gating kinetics, and V_hys_ of the *I*_K(erg)_ obtained in GH_3_ cells. However, the magnitude of the *I*_Na_ in these cells failed to be affected by this compound, while that of the *I*_K(DR)_ was mildly suppressed. We also observed that the *I*_K(erg)_ identified in MA-10 Leydig tumor cells was subjected to a block by isoPLB. The experimental observations prompted us to imply that the reduction by isoPLB or PLB of *I*_K(erg)_ could potentially contribute to their anti-neoplastic actions, presuming that similar in vivo findings occur.

Earlier investigations have demonstrated the effectiveness of isoPLB in perturbing mitochondrial respiration or the mitochondria-mediated induction of reactive oxygen species in many types of neoplastic cells [10,12,15,25,26]. Mitochondrial dysfunction would be expected to decrease cytosolic ATP content, thereby resulting in changes in the activity of ATP-sensitive K^+^ (K_ATP_) channels [74,81]. However, it seems that the inhibitory effect of isoPLB or PLB on the *I*_K(erg)_ magnitude seen in GH_3_ cells is unlikely to be mainly due to its activity as a regulator of K_ATP_ channels. The notable reasons can be explained as follows: (1) during whole-cell configuration in our experimental conditions, the ATP concentration of the pipette internal solution was 3 mM, a value that suffices to fully block the activity of K_ATP_ channels [74,75], and (2) an isoPLB-induced block of *I*_K(erg)_ could not be counteracted by subsequent exposure to diazoxide, an opener of K_ATP_ channels [74,75].

It needs to be mentioned herein that PLB was able to suppress the activities of intestinal Ca^2+^-activated Cl^−^ channels and the cystic fibrosis transmembrane conductance regulator (CFTR) or other types of Cl^−^ channels in cancer cells [82,83]. Pituitary cells have also reported the functional expression of Cl^−^ channels [30,64,84]. However, under our whole-cell current recordings for the measurement of *I*_K(erg)_, Cl^−^ ions inside the internal solution has been almost replaced with aspartate. Moreover, the *I*_K(erg)_ magnitude measured during long-lasting step hyperpolarization was not altered by adding chlorotoxin, an inhibitor of Cl^−^ channels [70], yet was sensitive to a block by azimilide, a blocker of *I*_K(erg)_ [71]. The block by isoPLB or PLB of *I*_K(erg)_ seen in GH_3_ cells is unlikely to be associated with its action on the activity of Cl^−^ channels reported recently [82,83].

In this study, as with azimilide, the presence of DTDP alone, a sulfhydryl oxidizing agent [72,73], was found to suppress *I*_K(erg)_ amplitude in GH_3_ cells. However, with cell exposure to DTDP, the subsequent addition of isoPLB failed to exert further suppressive effects on *I*_K(erg)_. PLB has been reported to induce free radical species in cervical cancer cells [3], as well as being an inhibitor of NADPH oxidase-4 [26], while 2-amino-1,4-naphthoquinone could induce the production of free radicals [56]. It is therefore plausible to speculate that the interaction with K_erg_ channels accompanied by the block of *I*_K(erg)_ could be shared to some extent by the DTDP and isoPLB molecules. To what extent the isoPLB-mediated block of *I*_K(erg)_ is associated with the production of reactive oxygen species also remains to be further resolved.

We have shown that *I*_K(erg)_ in GH_3_ cells did undergo V_hys_ behavior during the inverted isosceles-triangular V_ramp_, indicating that the K_erg_ channel residing in GH_3_ cells displays a clear V_hys_ in the voltage gating closely linked to the voltage-sensor domain residing in the channel [67,68]. The *I*_K(erg)_’s V_hys_ intrinsically in GH_3_ cells reflects that a mode shift during channel activation may exist, because the voltage sensitivity of the gating charge movement relies on the previous state (or conformation) of K_erg_ channels. Under such a scenario, as the membrane potential become negative (i.e., downward ramp of the double V_ramp_), the voltage dependence of K_erg_ channels may shift the mode of V_hys_ to one which occurs at more negative potentials, leading to an increase in membrane repolarization. However, as the membrane potential becomes depolarized (i.e., during initiation of action potentials or the upward limb of the triangular V_ramp_), the voltage dependence of *I*_K(erg)_ activation would switch to less depolarized voltages with a smaller current magnitude, thereby having the tendency to increase cell excitability. Both the slowing in recovery of the *I*_K(erg)_ block and the increase in the decay of *I*_K(erg)_ elicited with the envelope-of-tail method also strongly support the notion that the molecule of isoPLB or PLB interacts with the open states (conformations) of the K_erg_ channel. Under this scenario, the block by isoPLB or PLB of *I*_K(erg)_ demonstrated herein would be expected to have an important impact on the discharge patterns of action potentials occurring in excitable cells.

The resulting IC_50_ values required for an isoPLB-mediated block of peak and sustained *I*_K(erg)_ were 18.3 or 2.4 μM, respectively. From the first-order reaction scheme, the K_D_ value was quantitatively calculated to be 2.58 μM, a value that was noted to dovetail with the IC_50_ (i.e., 2.4 μM) for isoPLB-induced inhibition of sustained *I*_K(erg)_ evoked by long-lasting membrane hyperpolarization. An earlier study has shown that the isoPLB at a concentration of 2.5 μM could suppress the activity of mitochondrial respiration through a mechanism of its inhibitory action on complex IV activation [12]. Moreover, any modifications by isoPLB or PLB of *I*_K(erg)_ depend not only on the concentration of isoPLB or PLB given but also on different confounding variables, including the pre-existing level of the resting potential, various discharge patterns of action potentials, and their combinations, presuming that the magnitude of *I*_K(erg)_ is adequately and functionally expressed in the cells examined. Therefore, whatever the detailed mechanism of isoPLB-inhibited actions on *I*_K(erg)_ which remains resolved, the block by isoPLB or PLB of *I*_K(erg)_ demonstrated herein is of pharmacological relevance [35,62]. It is likely that the presence of isoPLB or PLB may induce the secretion of prolactin in vivo directly through the inhibition of *I*_K(erg)_ [30,31,36,46,55,58]. However, to what extent an isoPLB-mediated block of *I*_K(erg)_ affects cardiac function [32] remains to be delineated.

Since our results have shown that isoPLB and PLB have inhibitory effects on the magnitude, gating kinetics, and V_hys_ of *I*_K(erg)_, we next investigated whether isoPLB and PLB would directly bind to the hERG protein to affect its functional activity. The hERG protein was docked with isoPLB, PLB, and the known inhibitor doxazosin, through PyRx software. The predicted binding sites of these compounds were shown in Figure 10. Doxazosin forms hydrogen bonds with residue Y616 in pore helix P1 and forms hydrophobic contacts with residues V612, L615, F619, S620, T623, and F627 in pore helix P1 and residues Y652, and F656 in helix S6. IsoPLB forms hydrogen bonds with residue N629 in pore helix P1 and forms hydrophobic contact with residues V612, T613, and Y616 in pore helix P1 and residues L589, Q592, and I593 in the segment between S5 and P1. PLB form hydrophobic contacts with residues A565, and C566 in helix S5, residues T569, I607, K610, and Y611 in the segment between S5 and P1, and residue A614 in pore helix P1. Previous studies have reported that the expression of hERG channels in a variety of tumor cells could facilitate cell proliferation [47,48,85], and doxazosin inhibits hERG currents to reduce the proliferation of tumor cells [86,87]. Since our results have shown an inhibitory effect of isoPLB and PLB on the hERG current and a predicted interaction between the hERG protein and isoPLB or PLB, isoPLB and PLB could potentially be applied to treat neoplastic cells. Due to the fact that isoPLB binds preferentially to the extracellular domain of hERG, it is likely a more effective inhibitor than doxazosin.

In light of the present study, apart from the modifying effects of isoPLB on mitochondrial respiration [12], our results suggest that the inhibitory effects of isoPLB or PLB on the magnitude, gating kinetics, and V_hys_ of *I*_K(erg)_, appear to be obligate mechanisms. Through the ionic mechanism of their actions occurring in a concentration-, time-, state-, and hysteresis-dependent manner, it or other structurally similar compounds are anticipated to influence the functional activities or aberrant growth of different neoplastic cells, if similar in vivo results are found [9,16,29,48,49,50,51,54,82]. IsoPLB, PLB, or other structurally related compounds [88] could be intriguing compounds useful for characterizing the K_erg_ channels.

## Figures and Tables

**Figure 1 biomedicines-10-00780-f001:**
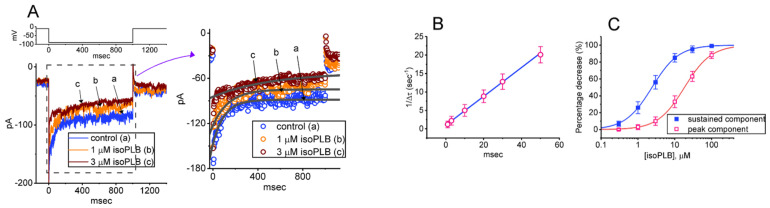
Effect of isoPLB on *I*_K(erg)_ identified from pituitary GH_3_ cells. These recordings were undertaken in cells immersed in a high-K^+^, Ca^2+^-free solution containing 1 μM tetrodotoxin (TTX) and 0.5 mM CdCl_2_, refilling the pipette that was used with K^+^-enriched internal solution. (**A**) Representative current traces evoked by 1-s step hyperpolarization (indicated in the upper part). a: control (i.e., absence of isoPLB), b: 1 μM isoPLB; c: 3 μM isoPLB. The right side of (**A**) indicates an expanded record shown in dashed box (left side). Current trajectory, the data points of which are indicated in open circles, was satisfactorily fitted with single exponential (in smooth gray line). In (**B**), a linear relationship of the resulting 1/∆*τ* value versus the isoPLB concentration was constructed and then plotted (mean ± SEM; *n* = 8 for each point). From the minimal reaction scheme (as elaborated under Materials and Methods), the forward (*k*_+1_*) and backward (*k*_−1_) rate constants for the isoPLB-mediated block of *I*_K(erg)_ in GH_3_ cells were computed as 0.39 s^−1^μM^−1^ and 1.01 s^−1^, respectively. (**C**) Concentration-dependent relationship of isoPLB on the peak (red open squares) or sustained (blue solid squares) components of *I*_K(erg)_ evoked by 1-s membrane hyperpolarization from −10 to −90 mV (mean ± SEM; *n* = 8 for each point). The smooth line indicates the goodness-of-fit to the Hill equation as described under Materials and Methods.

**Figure 2 biomedicines-10-00780-f002:**
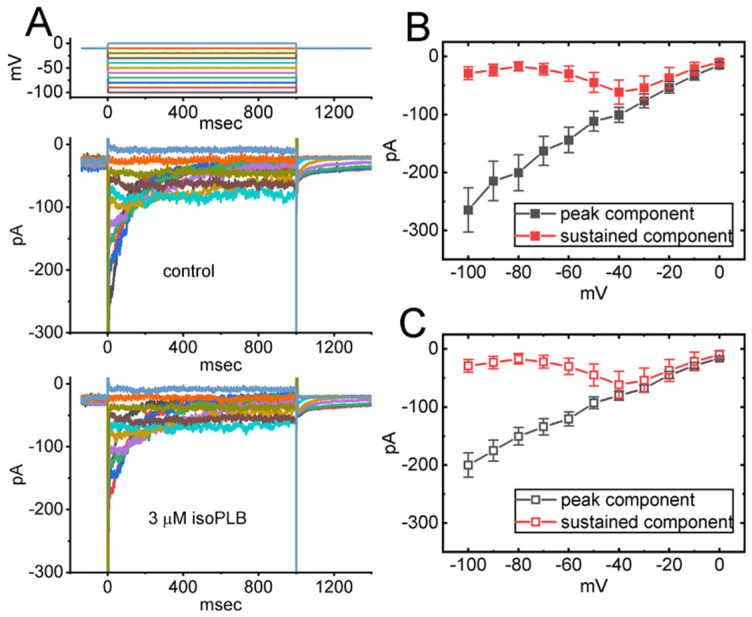
Inhibitory effect of isoPLB on steady-state *I–V* relationship of hyperpolarization-activated *I*_K(erg)_ identified in GH_3_ cells. In this series of experiments, we placed cells in a high-K^+^, Ca^2+^-free solution which contained 1 μM TTX and 0.5 mM CdCl_2_, and the pipette used for recordings was thereafter filled with a K^+^-enriched solution. We held the examined cell at −10 mV and a set of command potentials ranging between −100 and 0 mV with a duration of 1 s was imposed on it. (**A**) Superimposed current traces acquired in the control period (i.e., absence of isoPLB) (upper) and during cell exposure to 3 μM isoPLB (lower). The voltage-clamp protocol is indicated in the top part, and potential traces shown in different colors correspond with currents which were evoked by the same level of step command. In (**B**,**C**), average *I–V* relationships of the peak (black squares) or sustained component (red squares) of *I*_K(erg)_ attained in the absence (solid symbols) and presence (open symbols) of 3 μM isoPLB were demonstrated, respectively. The peak and sustained amplitudes of *I*_K(erg)_ were taken at the beginning and end-pulse of each step command applied. Each point represents the mean ± SEM (*n* = 7–8). The analyses with or without the addition of isoPLB among different levels of potentials given were performed in (**B**,**C**) with ANOVA-2 analysis (*p* < 0.05).

**Figure 3 biomedicines-10-00780-f003:**
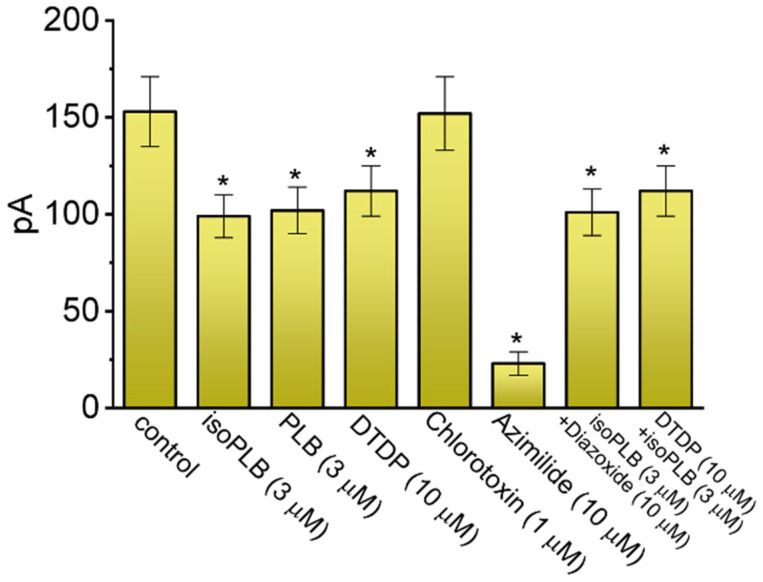
Comparison among effects of isoPLB, PLB, 4,4′-dithiopyridine (DTDP), chlorotoxin, azimilide, isoPLB plus diazoxide, and DTDP plus isoPLB on *I*_K(erg)_ recorded from GH_3_ cells_._ In these experiments, we placed cells in a high-K^+^, Ca^2+^-free solution containing 1 μM TTX and 0.5 mM CdCl_2_, and the recording electrode was filled up with K^+^-containing solution. The tested cell was held at −10 mV and a 1-s step hyperpolarization to −90 mV was subsequently applied to evoke *I*_K(erg)_. The current amplitude taken during cell exposure to different tested compounds was measured at the beginning of the hyperpolarizing step. Each bar represents the mean ± SEM (*n* = 7–8). Statistical analyses among different groups were performed with ANOVA-1 (*p* < 0.05). * Significantly different from control (*p* < 0.05). isoPLB: isoplumbagin; PLB: plumbagin; and DTDP: 4,4′-dithiodipyridine.

**Figure 4 biomedicines-10-00780-f004:**
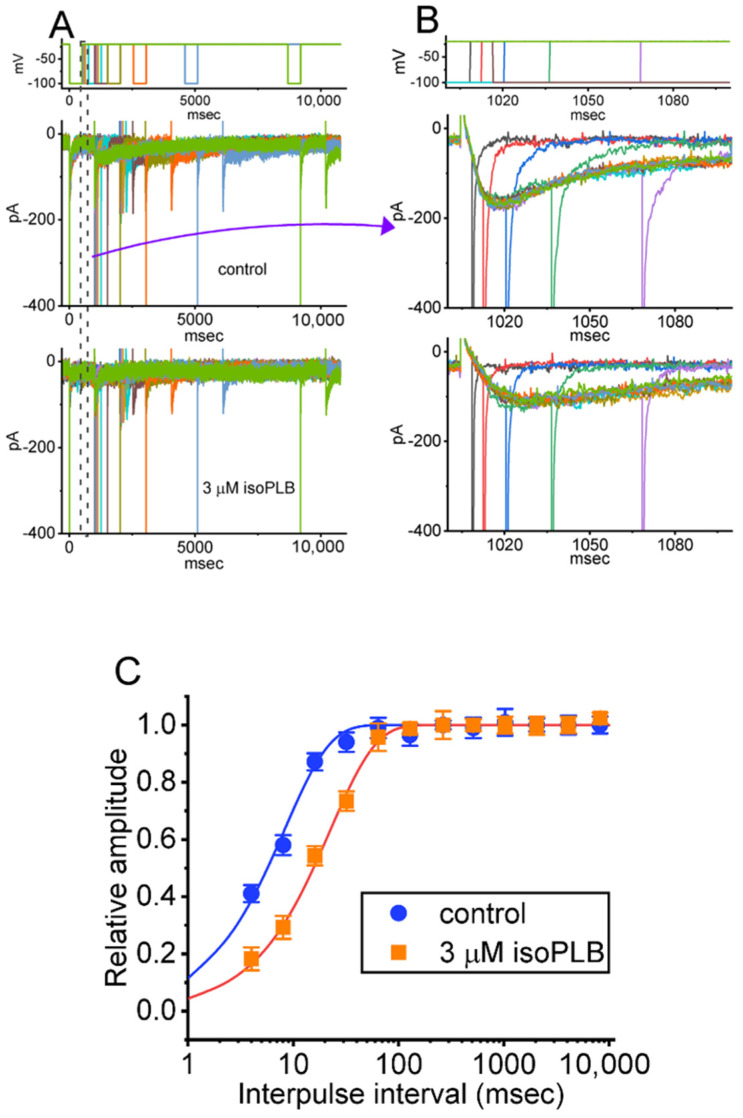
Modification by isoPLB of the recovery of the *I*_K(erg)_ block seen in GH_3_ cells. These measurements were undertaken in cells bathed in a high-K^+^, Ca^2+^-free solution, and the recording electrode was filled with a K^+^-containing solution. As the whole-cell configuration was firmly established, we imposed a two-step voltage protocol with a geometric progression onto the examined cells. (**A**) Superimposed current traces attained in the control period (i.e., absence of isoPLB) and during cell exposure to 3 μM isoPLB. The uppermost part in (**A**,**B**) shows the pulse protocol applied. In (**B**), the potential and current traces are the expanded records from the dashed box in (**A**). (**C**) Relationship of the interpulse interval versus the relative amplitude obtained in the absence (solid blue circles) and presence (solid orange squares) of 3 μM isoPLB (mean ± SEM; *n* = 8 for each point). The relative amplitude achieved in the ordinate was taken when the second amplitude evoked by a 1-s hyperpolarizing voltage-clamp command from −10 to −100 mV was divided by the first. The smooth curve without and with cell exposure to 3 μM isoPLB was least-squares fitted to a single exponential with time constants of 8.2 and 22.4 msec, respectively. The *y* axis is illustrated with a logarithmic scale. Of notice, the presence of isoPLB can lead to a prolongation of recovery from the *I*_K(erg)_ block evoked by such a two-step voltage command protocol.

**Figure 5 biomedicines-10-00780-f005:**
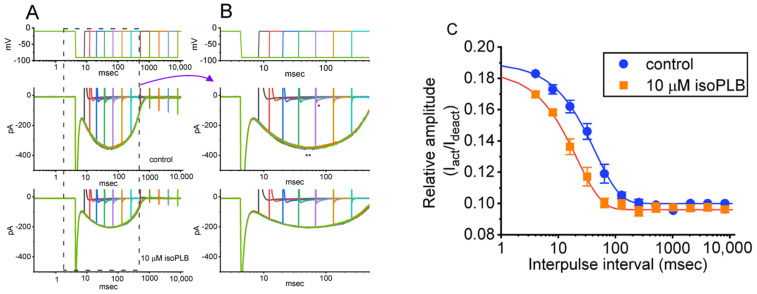
IsoPLB-mediated changes in *I*_K(erg)_ magnitude evoked by the envelope-of-tail test. Cells were bathed in a high-K^+^_,_ Ca^2+^-free solution and we then filled the pipette with a K^+^-enriched solution. Inward deactivating and activating currents were evoked by 4-ms to 8.192-s step hyperpolarizations to −90 mV with a geometric progression from a holding potential of −10 mV. (**A**) Representative potential and current traces evoked by the envelope-of-tail test. In (**B**), potential and current traces are the expanded records from the dashed box in (**A**). Of note, the *x* axis for the potential and current tracings appearing in (**A**,**B**) is scaled with a geometric progression. * indicates the *I*_act_ of hyperpolarization-evoked *I*_K(erg)_ and ** indicates the *I*_deact_ of the current. The uppermost part in (**A**,**B**) shows the voltage-clamp protocol delivered. (**C**) Relationship of the relative amplitude versus pulse duration obtained with or without the application of 10 μM isoPLB (mean ± SEM; *n* = 7 for each point). The relative amplitude appearing at the *y* axis was measured, when the inwardly decaying current (i.e., *I*_act_) following return to −10 mV was divided by peak *I*_K(erg)_ (i.e., *I*_deact_) in response to membrane hyperpolarizing pulses with varying pulses in a geometric progression. Of note, the relationship is illustrated with a semi-logarithmic plot. The smooth curve without or with the addition of 10 μM isoPLB indicates the best fit to single exponential function with a time constant of 44.3 or 21.2 msec, respectively. The decaying rate of *I*_K(erg)_ evoked during the envelope-of-tail occurred in single exponential function, and the presence of isoPLB yielded an increase in the decay rate of *I*_K(erg)_ evoked upon such envelope-of-tail test.

**Figure 6 biomedicines-10-00780-f006:**
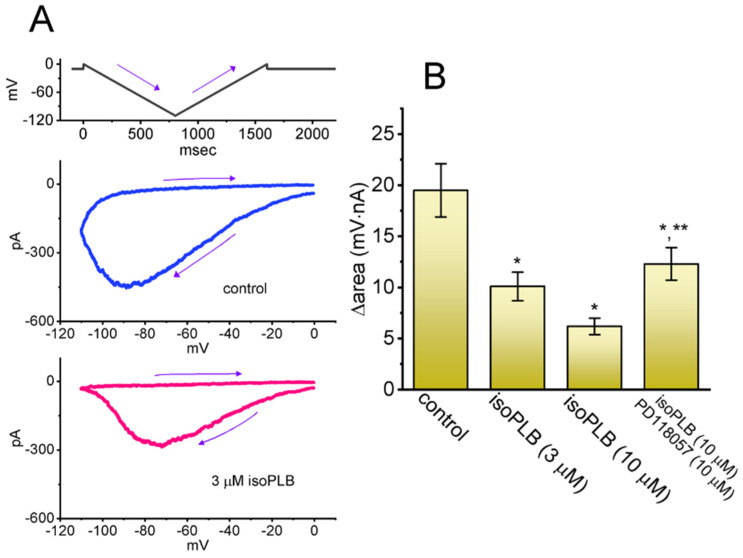
Effect of isoPLB on the V_hys_ of *I*_K(erg)_ evoked during the double (i.e., inverted isosceles-triangular) ramp voltage (V_ramp_). These experiments were conducted in cells which were bathed in a Ca^2+^-free Tyrode’s solution containing 1 μM TTX and 0.5 mM CdCl_2_; we then filled the pipette with a K^+^-enriched solution. (**A**) Representative V_hys_ traces (i.e., the relationship of descending or ascending *I*_K(erg)_ versus membrane voltage) obtained in the control period (upper, blue color) (i.e., absence of isoPLB) and during exposure to 3 μM isoPLB (lower, red color). The top part indicates the voltage-clamp protocol delivered. The arrow in each panel indicates the direction of *I*_K(erg)_ over time during the elicitation of an inverted isosceles-triangular V_ramp_. (**B**) Summary bar graph showing the effectiveness of isoPLB and isoPLB plus PD118057 in altering the V_hys_ ∆area (i.e., area under the curve activated during the downward and upward limb of triangular V_ramp_). The ramp speed in these experiments was set at ±183.3 mV/sec. Each bar indicates the mean ± SEM (*n* = 8). The analysis was carried out with ANOVA-1 (*p* < 0.05). * Significantly different from control (*p* < 0.05) and ** significantly different from the group of isoPLB (10 μM) alone (*p* < 0.05).

**Figure 7 biomedicines-10-00780-f007:**
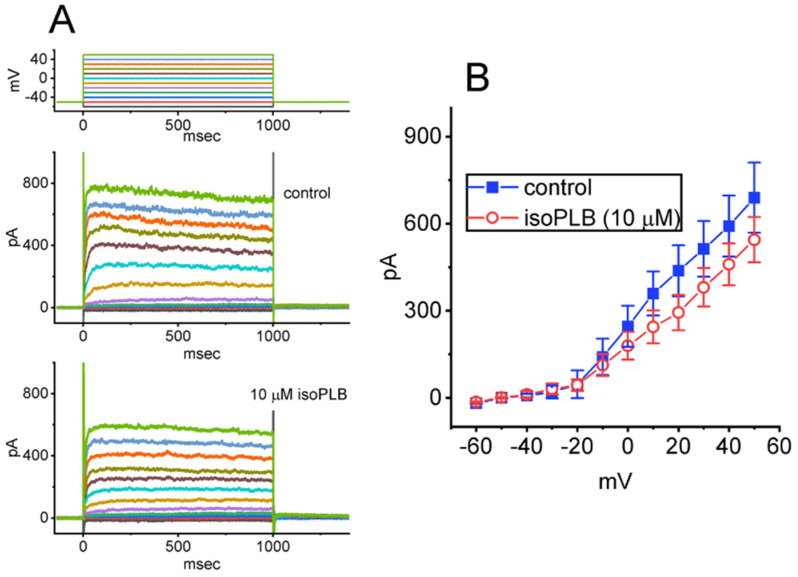
Inhibitory effect of isoPLB on delayed-rectifier K^+^ current (*I*_K(DR)_) inherently in GH_3_ cells. The experiments were performed as cells were bathed in a Ca^2+^-free Tyrode’s solution which contained 1 μM TTX and 0.5 mM CdCl_2_, while the recording pipette was filled with a K^+^-enriched solution. The examined cell was maintained at −50 mV and we imposed a series of 1-s step commands ranging between −60 and +50 mV in 10-mV step. (**A**) Representative current traces obtained in the control period (i.e., absence of isoPLB) (upper) and during cell exposure to 10 μM isoPLB (lower). The uppermost part shows the voltage-clamp protocol used. (**B**) Average *I–V* relationships of *I*_K(DR)_ taken in the absence (solid blue squares) and presence of 10 μM isoPLB (open red circles) (mean ± SEM; *n* = 7 for each point). We measured the current amplitude at the end of each step voltage-clamp command. Of note, cell exposure to 10 μM isoPLB results in the mild inhibition of the *I*_K(DR)_ amplitude measured over the entire voltage-clamp range imposed. The analyses in the absence and presence of isoPLB among different levels of membrane potentials were performed with ANOVA-1 (*p* < 0.05).

**Figure 8 biomedicines-10-00780-f008:**
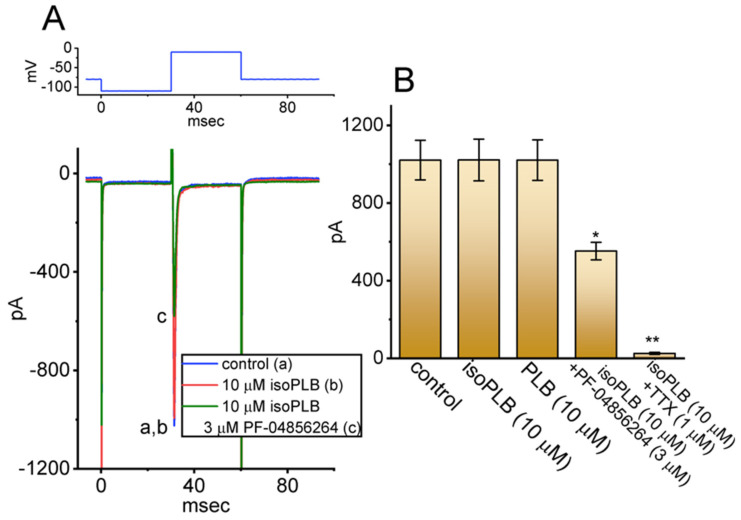
Effect on isoPLB on *I*_Na_ in GH_3_ cells. We bathed cells in a Ca^2+^-free Tyrode’s solution containing 0.5 mM CdCl_2_ and 10 mM TEA, and the pipette was filled up with a Cs^+^-containing solution. (**A**) Original current traces obtained in the absence (a) and presence of 10 μM isoPLB (b) or 10 μM isoPLB plus 3 μM PF-04856264. The top part depicts the voltage protocol applied to the tested cell. (**B**) Summary bar graph showing the effect of isoPLB, PLB, isoPLB plus PF-04856264, and isoPLB plus TTX on the peak amplitude of *I*_Na_ during brief depolarizing pulse (30 ms in duration) from −100 to −10 mV (mean ± SEM; *n* = 7 for each bar). * Significantly different from control (*p* < 0.05) and ** significantly different from control (*p* < 0.01). The analysis among different groups was performed with ANOVA-1 (*p* < 0.05).

**Figure 9 biomedicines-10-00780-f009:**
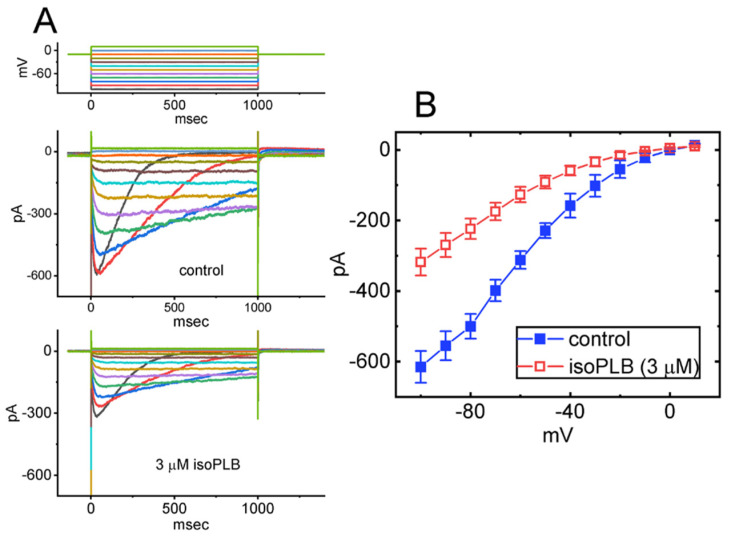
Effect of isoPLB on *I*_K(erg)_ obtained in MA-10 Leydig tumor cells. Similar to the experiments above conducted in GH_3_ cells, we immersed cells in a high-K^+^, Ca^2+^-free solution, and we filled the recording pipette with a K^+^-enriched solution. (**A**) Superimposed current traces obtained in the control (upper) and during MA-10-cell exposure to 3 μM isoPLB (lower). The *I*_K(erg)_ was evoked by the voltage-clamp protocol (indicated in the top part). (**B**) Average *I–V* relationship of the peak component of *I*_K(erg)_ in the absence (solid blue squares) and presence (open red squares) of 3 μM isoPLB measured from MA-10 cells (mean ± SEM; *n* = 7). We measured the current amplitude at the start of each voltage pulse. The analyses with or without the addition of isoPLB measured at different levels of membrane potentials were performed with ANOVA-2 (*p* < 0.05).

**Figure 10 biomedicines-10-00780-f010:**
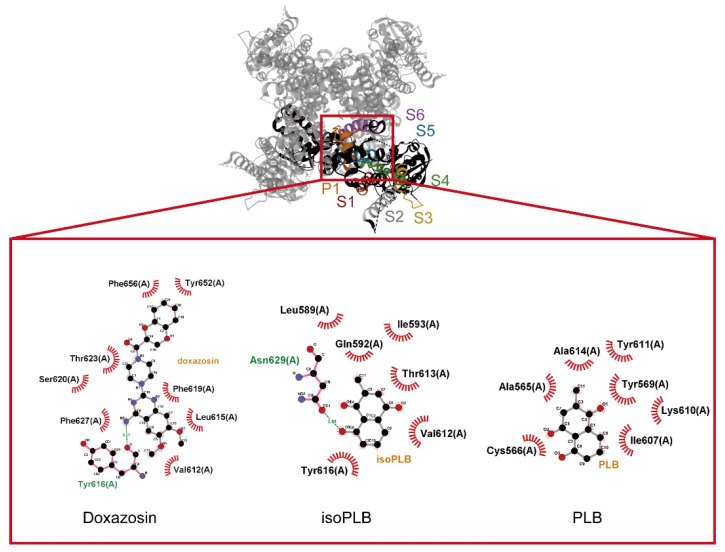
Docking results of hERG and doxazosin, isoPLB, and PLB. The protein structure of hERG was obtained from PDB (PDB ID: 5VA1) and the compound structure of doxazosin, isoPLB, and PLB were obtained from PubChem (compound CID: 3157, 10205, 375105). The structure of hERG amino acid 520–660 was extracted through PyMOL and then docked with three compounds through PyRx. The diagrams of interaction between hERG and compounds were generated by LigPlot+. The red arcs with spokes radiating toward the ligand represent hydrophobic contacts and green dotted lines represent hydrogen bonds.

## Data Availability

The original data are available upon reasonable request to the corresponding author.

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
