# Peer review of "The Effectiveness of Isoplumbagin and Plumbagin in Regulating Amplitude, Gating Kinetics, and Voltage-Dependent Hysteresis of erg-mediated K+ Currents"

_biomedicines, 2022, doi:10.3390/biomedicines10040780_

Round 1

Reviewer 1 Report

The authors studied the influence of isoplumbagin and plumbagin on membrane ion channels in detail. This study is well designed and the conclusion is solid based on their results.  The paper is written relatively well. However, I feel there are too many long sentences were used, which makes it a bit difficult to read and understand. A few mistakes were spotted.

Line 201, yield to yielded

Line 335, hyperpolarizaing to  hyperpolarizing

Line 343, significant to significantly

Line 435 and 486, level to levels

In addition, I am not very much convinced with the "High effectiveness" in the title, probably the authors can think about whether this should be modified.

Reviewer 2 Report

The Authors in this article describe the effectiveness of isoplumbagin (isoPLB) and plumbagin (PLB) on erg-mediated K+ current, in rat pituitary tumor cell line (GH3). The main result, confirmed by a second cell tumor line, MA-10 Leydig cells, is that isoPLB and PLB could depress the magnitude of K(erg) current, in a concentration-, time-, state-, and hysteresis-dependent manner.

General consideration

The authors suggest here (493: The experimental observations prompted us to imply that the reduction by isoPLB or PLB of IK(erg) could potentially contribute to its anti-neoplastic actions) and somewhere else that the is a relationship between the effect on IK(erg) currents and the capability of influencing functional activities or aberrant growth of neoplastic cells, indicating as the starting mechanism for the cascade signaling that drive cells behavior,  the depressing effect on this channels.

The patch-clamp technique used by the group in this work and in others allows them to investigate the effect of a substance on the activity of a channel protein on which a specific binding site is present. The interaction can activate, modulate or block the entry of the ionic species for which that channel is selective. This type of studies allows to obtain very detailed information on a specific phenomenon but partial with respect to a more clinical vision. To formulate a hypothesis that can be translated to in vivo conditions, the activities of potential drugs acting at the level of ion channels, in my opinion, should also be tested with different experimental approaches, always in vitro but more systemic, such as cell survival/proliferation or the study of molecular pathways triggered by channel activity. My suggestion is to create a collaboration with those who use other methods of analysis or otherwise, to collect all available literature that supports a translational hypothesis (eg: potential drug on pituitary tumor) and argue according to previous and current evidence in a clear and detailed way.

31: Alternatively, the IK(erg) identified in MA-10 Leydig tumor cells was also blocked by the presence of isoPLB

Here and elsewhere has been used the term blocked which is not correct since the effect is actually a drastic and not a complete reduction of the peak amplitude of IK(erg).

What is the origin of GH3 ???

I would like the cell lines to be better described and justified

46: ….., another hydroxyl-1,4-napththoquinone, is another an alkaloid obtained from the…

Remove the second “another” (red)

49: against an array of cancer cells, including oral or tongue squamous cell carcinoma,

Remove the “cells” (red)

51: Alternatively, PLB was shown to induce apoptotic changes in lung.

Replace with: Nevertheless PLB was also shown….

Besides from these few samples the entire text needs a careful linguistic revision and formatting

103: while other chemicals, such as aspartic acid, CdCl2, CsCl, CsOH, EGTA and HEPES, were of laboratory grade.

Better to name the production company

If "keeping the cell at -10 mV" means keeping a holding potential of -10 mV would it be better to specify it in the methods?

107-116: The manufacturer should be added to the list of chemicals present in the solutions.

133: The microscope was coupled to a video camera system with magnification up to 1500×, in order to monitor cell size….

1500x of magnification for patch clamp isn't too much, and usually the order of magnification is referred to the objectives.

135: Cells were immersed at room temperature (20-25 °C) in HEPES-buffered normal Tyrode’s solution containing 1.8 mM CaCl2, the composition of which is stated above.

I’m confused, in the rest of the experimental protocol descriptions it is stated that the Tyrode solution is calcium free….

Figure 1

Missing description Fig. 1 c)

249: From the minimal binding scheme (as elaborated under Materials and Methods), the forward (k+1*) and backward (k-1) rate constants for isoPLB-mediated block of IK(erg) in GH3 cells were computed to 0.39 sec-1mM-1 and 1.01 sec-1, respectively.

Wouldn't it be better to put it in the results instead of in the caption and refer to the figure?

The traces of the currents are very thick and noisy (in particular in Figure 1 and 4), so it is difficult to distinguish them in the figures and follow their progress.

297: In these experiments, cells were bathed in high-K+, Ca2+-free solution, and the pipette that we used was filled up with K+-containing solution.

It means there are no Na and Cl blockers right? Therefore it is not correct to speak of the Kerg current if it has not been isolated. And this is always the case. It is extremely important to first isolate the current you want to study and then study the effects of substances on it by subtraction. Although it was done in a previous publication (Hsu, HT, Lo, YC, & Wu, SN (2020). International journal of molecular sciences, 21 (4), 1441) I find that it is necessary to show also in this the trace with blocking or at least describe the result and cite the source.

300: Current amplitude was measured at the start of  the hyperpolarizing command pulse during cell exposure to different tested compounds.

Doesn't mean much… what was actually measured ???

303: ……with ANOVA-1 (P < 0.05). *Significantly different from control (P < 0.05). isoPLB: isoplumbagin; PLB: plumbagin; DTDP: 4,4’-dithiodipyridine

Maybe something is missing at the end of this sentence?

The bath and pipette solutions are not shown in figure 4.
